Distribution and phylogeography of the genus Mattirolomyces with a focus on the Asian M. terfezioides haplotypes

Wei Jie 1
http://orcid.org/0000-0003-4035-8587 Grebenc Tine 2
Zhang Xuan 3
Xiang SiMin 3
Fan Yongjun 4 5 fanyj1975@163.com
1 Forestry College, Inner Mongolia Agricultural University China , Huhhot , China
2 Slovenian Forestry Institute , Ljubljana , Slovenia
3 Baotou Teachers College, Inner Mongolia University of Science and Technology, China , Baotou , China
4 Inner Mongolia Key Laboratory for Biomass-Energy Conversion , Baotou , China
5 School of Life Science and Technology, Inner Mongolia University of Science and Technology , Baotou , China
Sosa Victoria
Electronic publication date: 2022 Jul 26
Publication date: 2022
Volume: 10
Electronic Location ID: e13511
Received 2021 Nov 18; Accepted 2022 May 6
Copyright: © 2022 Wei et al.
Copyright year: 2022
Copyright holder: Wei et al.
License: This is an open access article distributed under the terms of the Creative Commons Attribution License, which permits unrestricted use, distribution, reproduction and adaptation in any medium and for any purpose provided that it is properly attributed. For attribution, the original author(s), title, publication source (PeerJ) and either DOI or URL of the article must be cited.
License URL: https://creativecommons.org/licenses/by/4.0/

Keywords: Mattirolomyces terfezioides, Desert truffle, Inner Mongolia, Phylogeography

Funding: Natural Science Foundation of Inner Mongolia 2020MS03001 Educational Commission of Inner Mongolia NJYT-18-A21 Science and Technology Project of Inner Mongolia 2019GG002 Inner Mongolia Agricultural University Research Project J4-1766 “Methodology approaches in genome-based diversity and ecological plasticity study of truffles from their natural distribution areas” Research Program in Forest Biology, Ecology and Technology (P4-0107) The study was financially supported by the Natural Science Foundation of Inner Mongolia (No. 2020MS03001), the Educational Commission of Inner Mongolia (No. NJYT-18-A21), the Science and technology project of Inner Mongolia (No. 2019GG002), and the Research fund for young teachers of College of Forestry, Inner Mongolia Agricultural University. Slovenian Partner was co-financed by the research project J4-1766 “Methodology approaches in genome-based diversity and ecological plasticity study of truffles from their natural distribution areas” and the Research Program in Forest Biology, Ecology and Technology (P4-0107) of the Slovenian Research Agency. We thank Dr. Xiaojuan Huang for meteorological data collection. The funders had no role in study design, data collection and analysis, decision to publish, or preparation of the manuscript.

==============================
Mattirolomyces is an edible commercial sequestrate genus that is globally distributed. From the five described taxa of this genus, Mattirolomyces terfezioides is the most common species in Asia. Our recent attempts to locate M. terfezioides outside its current distribution area in China documented its first records in areas of poplar trees with the lowest known temperature and precipitation averages ever recorded for this species. This peculiar ecology was not reflected on the species-morphological features nor on its phylogenetic position in the genus. The first attempt to apply the phylogenetic network approach to Mattirolomyces revealed its geographic origin in the Asian-Pacific areas prior to frequent long-distance migration events. Based on data from recent study areas, we found that the collections from Inner Mongolia and the Shanxi province were similar to European collections. Asian haplotypes were less distant from the outgroup comparing to collections from Europe, supporting the hypothesis that M. terfezioides was originated from this Chinese area and was subsequently transported to Europe. Exploring M. terfezioides ecology and its mycorrhiza potential to grow in association with poplars would be of great importance for planning cultivation projects of this valuable desert truffle species in Central and Eastern China, a currently underexploited economic sector that deserves further ecological and M. terfezioides mycorrhizal synthesis investigations.

Introduction

The genus Mattirolomyces (Tuberaceae, Pezizales) was called the “Mattirolo fungus” by Fisher in 1938 after Mattirolo which was the first to describe the type species of this genus. This last species Mattirolo named in 1887 as Choiromyces terfezioides (Fischer, 1938), however, Moreno, Alvarado & Manjón (2012) confirmed later its separate generic position from the genus Choiromyces. The type specimen was firstly collected from clay agricultural soils in a non-typical ecological location in Piemonte, Northern Italy, and was considered at that time as a potential symbiotic partner of Prunus avium (Mattirolo, 1887). The Mattirolomyces taxon belongs to the ascomycetous desert truffles. All known species of this genus form sequestrate to hypogeous sporocarps (Fischer, 1938). Currently, five species in the genus Mattirolomyces have been shown to have a wide geographical distribution, commonly collected from areas with low and variable average rainfall and high summer temperatures (Kagan-Zur et al., 2014). The type species of the genus Mattirolomyces terfezioides (Mattir.) E. Fisch was recorded in Europe and Asia (Fischer, 1938); Mattirolomyces spinosus (Harkn.) Kovács, Trappe & Alsheikh was collected in North America and Pakistan (Kovács et al., 2011); Mattirolomyces mulpu Kovács, Trappe & Claridge was reported from Australia (Trappe, Kovács & Claridge, 2010); Mattirolomyces austroafricanus (Marasas & Trappe) Kovács was found from South Africa (Trappe, Kovács & Claridge, 2010); and Mattirolomyces mexicanus Kovács, Trappe & Claridge was described in Mexico (Kovács et al., 2011). The distribution pattern of these five species suggests a wide geographical (presumably global) range of the genus (Kagan-Zur et al., 2014) with most records coming from Eastern and Southeastern Europe (Glejdura & Kunca, 2012; Kagan-Zur et al., 2014; Assyov & Slavova, 2016).

The desert regions of the Southern Hemisphere have low and variable average rainfall and high summer temperatures. Mattirolomyces spp. and other desert truffles species have a long history of regular hunt and harvest for human consumption since prehistoric times (Trappe, Kovács & Claridge, 2010). However, M. terfezioides is rarely recognized as a valuable commercial species in Europe and Asia (Boa, 2004). Mattirolomyces spp. sporocarps are traditionally collected, sold, and consumed under the name Terfezia terfezioides (Mattir.) Trappe (Trappe, 1971), a synonym of Mattirolomyces terfezioides. Despite their great values as mycorrhizal species and culinary delicacy, this species is not well-known in the Northern Hemisphere (Kovács, Jakucs & Bagi, 2007). Most available collections of M. terfezioides are from Hungary (Europe) and Northern China, namely Beijing, Hebei Province, and Shanxi Province. Most of the Chinese collections date back several decades, with the most recent dating to 1986 (Wang, Liu & Sun, 2017). The economic and culinary value of the Chinese M. terfezioides collections have not yet been evaluated. Furthermore, this species has been considered long-lost in China by many mycologists.

We were motivated by a recently discovered M. terfezioides collection from the desert areas of Inner Mongolia, China to revive the study of the Chinese genus, characterize the current ecological span of the species, and prepare a morphology-based description of the Chinese collection. Due to a low number of records and available nucleotide sequences, no phylogeographic insight into the genus is currently available. Therefore, we aimed to position the Chinese collections of M. terfezioides in a phylogenetic network of the whole genus, focusing on the relationship between the Chinese collections and the collections from other areas worldwide, in order to ultimately hypothesize the genus’s geographic origin.

Materials and Methods

Study site and sampling

The most temperate continental part of China, Inner Mongolia, has a cold semi-arid (BSk) to cold desert (BWk) climate (Peel, Finlayson & Mcmahon, 2007). Although the occurrence of Mattirolomyces was not previously recorded in this area, the genus was found in most of its neighboring provinces, which indicated its potential for fruiting in Inner Mongolia as well. Sporocarps were primarily sought for in ecosystems that were suitable for Mattirolomyces (Kagan-Zur & Akyuz, 2014). When selecting sampling microlocations, we targeted the known ectomycorrhizal partner sites with pines and black locusts (Kagan-Zur & Akyuz, 2014), as well as young plantations of Populus alba L. ssp. pyramidalis (Bunge) W.Wettst. at various locations in the Baotou area, Inner Mongolia, China, between September and October 2018 and 2019.

The main climatic characteristics of the broader area where M. terfezioides was repeatedly collected were: an elevation of about 1,070 m in the surveyed area, average temperature of around 8.5 °C, lowest temperature of minus 27.6 °C, and highest temperature of 40.4 °C. The average annual rainfall in this area over the last 18 years is 301.6 mm (the minimum rainfall was 175.9 mm in 2005, and the maximum rainfall was 465.2 mm in 2003). The average annual rainfall in 2018 and 2019 was 364.8 and 327.6 mm, respectively, according to data from Inner Mongolia Meteorology. The soils are sandy, hyphal aggregates connected the roots with above ground parts of the plants, and the sporocarps of this fungus developed from the hyphal aggregates.

Sporocarps were collected by racking the soils following procedure in Castellano, Trappe & Luoma (2004). All sporocarps were photographed in situ with a Canon EOS 60D camera (Canon, Tokyo Japan), then dried in a forced-air dryer and kept in the Herbarium and Fungarium of Baotou Teachers’s College under accession number Fan0273.

Determining soil physical and chemical properties

The soil samples were taken from the immediate vicinity of the fruiting sporocarps to a depth of 10 cm. Soil pH was measured in 1M KCl (1;5 w/v). Organic C and total N were analyzed using the CHNS-analyzer system (Elementar Analysen Systeme GmbH, Hanau, Germany) with the burning method at 450 °C and 1,250 °C, respectively (Liu et al., 2012). We determined total organic matter, available phosphorus content, and available potassium as well as the total content of water-soluble salts following the standardized operation procedures (Pansu & Gautheyrou, 2007).

Morphological observation

The macro-morphological characterization of ascomata was performed using a stereomicroscope (Motic K400; Motic Asia, Kowloon, Hong Kong) following the Mattirolomyces morphological characters description of Kovács et al. (2011). The micro-morphological features were determined on 30 spores and asci using a light microscope (Motic BA410E with a Moticam2506 camera; Motic Asia, Kowloon, Hong Kong). Melzer’s reagent and Cotton blue chemical reactions were also used in order to improve the morphological identification. The spore morphology and ornamentation were examined using scanning electron microscopy (SEM).The observations were performed on a desiccated spore suspension coated with platinum-palladium using a vacuum metallizing machine (Hitachi E-1010; Hitachi, Tokyo, Japan). Electron microscope images were obtained with a Hitachi S-530 (Hitachi, Tokyo, Japan) scanning electron microscope.

DNA extraction, PCR amplification, and sequencing

DNA extraction, PCR amplification of the complete rDNA ITS region using primers ITS1f/ITS4 (White et al., 1990) and Taq PCR Master Mix (Biobasic, Markham, ON, Canada) as well as sequencing were carried out according to Wang, Liu & Sun (2017). PCR products were purified and sequenced at the Chengdu Institute of Biology, Chinese Academy of Sciences, Chengdu, Sichuan, China. The rDNA ITS sequence obtained in the present study was deposited in GenBank under the accession number listed in Table 1.

Table 1 Collected information of Mattirolomyces terfezioides nuclear rDNA ITS sequence generated in the present work and ITS sequences from the genus Mattirolomyces used in this study retrieved from GenBank or UNITE databases.

Species	GenBank accession numbers	Geographic origin	Climate (Köppen-Geiger climate classification)	Potential(*) symbiotic partners	Sequence reference	
Mattirolomyces terfezioides	KT963177	China, Hebei Province, Wanxian	Dwa	Robinia pseudoacacia	Wang, Liu & Sun (2017)	
Mattirolomyces terfezioides	KT963175	China, Beijing City	Dwa	Robinia pseudoacacia	Wang, Liu & Sun (2017)	
Mattirolomyces terfezioides	AJ305170	Italy, Ravenna	Cfa	n/a	Kovács et al. (2001)	
Mattirolomyces terfezioides	AJ305169	Hungary, Great Hungarian Plain, Kunfehértó	Cfb	n/a	Kovács et al. (2001)	
Mattirolomyces terfezioides	AJ272442	Hungary, Great Hungarian Plain, Őrbottyán	Dfa/Dfb	n/a	Kovács et al. (2001)	
Mattirolomyces terfezioides	AJ305045	Hungary, Great Hungarian Plain, Mogyoród	Dfa/Dfb	n/a	Kovács et al. (2001)	
Mattirolomyces terfezioides	AJ272443	Hungary, Great Hungarian Plain, Gyál	Dfa/Dfb	n/a	Kovács et al. (2001)	
Mattirolomyces terfezioides	AJ306556	Hungary, Great Hungarian Plain, Kunfehértó	Cfb	n/a	Kovács et al. (2001)	
Mattirolomyces terfezioides	AJ272444	Hungary, Great Hungarian Plain, Őrbottyán	Dfa/Dfb	n/a	Kovács et al. (2001)	
Mattirolomyces terfezioides	AF276681	Hungary, Surány	Dfa	n/a	Díez, Manjon & Martin (2002)	
Mattirolomyces terfezioides	AJ272445	Hungary, Sülysáp	Dfa/Dfb	n/a	Kovács et al. (2001)	
Mattirolomyces terfezioides	GQ231754	France, Provence-Alpes-Côte d’Azur, Le Thor	Csa	n/a	Trappe, Kovács & Claridge (2010)	
Mattirolomyces terfezioides	AJ306555	Hungary, Great Hungarian Plain, Kunfehértó	Cfb	n/a	Kovács et al. (2001)	
Mattirolomyces terfezioides	AF276680	Hungary, Csomád	Dfa	n/a	Kovács et al. (2001)	
Mattirolomyces terfezioides	KT025693	South Korea, Buk-myeon, Taean-gun	Dwa	Robinia pseudoacacia	Ka et al. (2015)	
Mattirolomyces terfezioides	KT963176	China, Shanxi Province, Taiyuan	BSk	Robinia pseudoacacia	Wang, Liu & Sun (2017)	
Mattirolomyces terfezioides	JF908728	Italy	n/a	n/a	Osmundson et al. (2013)	
Mattirolomyces terfezioides	AJ875015	Hungary	Dfa/Dfb	Robinia pseudoacacia	Bratek et al. (1996)	
Mattirolomyces terfezioides	KT963178	China, Shanxi Province, Taiyuan	BSk	Robinia pseudoacacia	Wang, Liu & Sun (2017)	
Mattirolomyces terfezioides	AJ875016	Hungary	Dfa/Dfb	Robinia pseudoacacia	Bratek et al. (1996)	
Mattirolomyces terfezioides	MN619773	China, Inner Mongolia, Baotou	BSk	Populus alba L. ssp. pyramidalis	this study	
Mattirolomyces spinosus	HQ660384	Pakistan, Punjab, Sheikhupura	BSh	n/a	Kovács et al. (2011)	
Mattirolomyces mexicanus	HQ660378	Mexico, Nuevo Leon, Guadalupe	BSh	n/a	Kovács et al. (2011)	
Mattirolomyces spinosus	HQ660381	USA, Louisiana, Natchitoches	Cfa	n/a	Kovács et al. (2011)	
Mattirolomyces austroafricanus	GQ231752	South Africa, Northern Cape province, Barkly West	BSh	n/a	Trappe, Kovács & Claridge (2010)	
Mattirolomyces mulpu	GQ231739	Australia, Northern Territory	Bwh	n/a	Trappe, Kovács & Claridge (2010)	
Elderia arenivaga(outgroup)	GQ231736	Australia, Northern Territory, Alice Springs Desert Park	Bwh	n/a	Trappe, Kovács & Claridge (2010)	
Elderia arenivaga(outgroup)	GQ231733	Australia, South Australia, Great Victoria Desert	Bwh/Bwk	n/a	Trappe, Kovács & Claridge (2010)	
Notes:

n/a, not available.

Species names and GenBank accession numbers were supplemented with geographic origin of the collection, climate, potential host plants and other site data, if available.

Phylogenetic analyses

Available and compete nuclear rDNA ITS sequences from the genus Mattirolomyces were retrieved from GenBank (Benson et al., 2013) and UNITE databases (Kõljalg et al., 2013) on December 12, 2019. We conducted a nucleotide search using the basic local alignment search tool (BLAST) with our representative sequence to found additional sequences that were potentially misnamed in the searched databases. A local Mattirolomyces spp. nuclear rDNA ITS sequence database was built based on available and reliable sequences, and we selected environmental parameters from the corresponding original papers of the sequences or directly from online databases (Table 1), with meteorological data from the latest FLUXNET synthesis dataset, the FLUXNET2015 database (http://fluxnet.fluxdata.org/data/fluxnet2015-dataset/data-processing/).

DNA sequences were assembled in BioEdit v5.0.9 (https://bioedit.software.informer.com). MEGA7.0 software (Kumar, Stecher & Tamura, 2016) was then used for multiple sequence alignment and phylogenetic analysis. The internal MEGA7 plug-ins were used for sequence alignment (ClustalW), testing for the best nucleotide substitution model (Model Test), maximum likelihood phylogenetic analysis (ML phylogenetic analysis), and construction of the phylogenetic tree. Kimura’s 2-parametric model was selected as the best model for a distance calculation of a given dataset. In order to evaluate the stability of the ML evolutionary tree topology, 1,000 bootstrap repetitions were run. Using the Bayesian method, we calculated Bayesian inference with MrBayes v. 3.1.2 (Ronquist & Huelsenbeck, 2003) and an HKY+G model. Four Markov chains were run for two runs from random starting trees for 1 million generations, until the split deviation frequency value <0.01. Every 100th generation was sampled. The Bayesian inference tree was visualized in FigTree 1.4.2. Branches that received bootstrap from ML ≥60% and Bayesian posterior probabilities (BPP) ≥0.95 were considered significantly supported.

For the phylogenetic network analysis, the same nucleotide dataset was realigned in MAFFT v. 7.304b (Katoh & Standley, 2013) and analyzed with a Median joining approach in Network5 (Bandelt, Forster & Röhl, 1999). The phylogenetic network constructed in Network5 was modified and annotated in the GNU General Public License program Inkscape 0.91 (https://inkscape.org/release/inkscape-0.91/).

Results

Taxonomy

Over 2 years of hunting for hypogeous fungi in Mattirolomyces-like habitats, we collected two independent collections made up of a total of 32 sporocarps, all from Populus alba ssp. pyramidalis plantations.

The morphological description of the collections from Inner Mongolia affiliate them to Mattirolomyces terfezioides (Mattir.) E. Fisch., as described by Fischer (1938). Ascomata (fresh specimens, Fig. 1A) were hypogeous or subepigeous, 4–5 cm in diam., subglobose to irregular massy, white, surface smooth to scabrous, lobed and furrowed; gleba solid, firm with minute pockets, white with narrow white veins (Fig. 1C). Taste and odor were strongly sweet when fresh. Dark brown nombril (0.5–1.0 cm in diam.) was found in some specimens as hyphal aggregates, attached with the base of the sporocarps (Fig. 1B). Paraphyses were absent.

Figure 1 Macro-and micromorphological characteristics of Mattirolomyces terfezioides.

(A) Ascocarps in situ; (B) nombril attached to the ascocarp; (C) gleba; (D, E) asci and spores; (F) spore with warty, blunt spines (scanning electron micrograph).

Microscopic features: peridium thin, 120–280 mm thickness, not differentiated from the gleba, composed of inflated hyphae and irregular, hyaline cells; gleba composed of interwoven septate hyphae 7.5–11(20) μm broad, with some free hyphal ends; asci randomly arranged in gleba, 8- or 10-spored, hyaline, globose to ellipsoid, pockety, saccate, cylindrical or clavate, (55) 65–95 (117) × (26) 35–45 (60) μm, sessile or occasionally sub-stipitate with a short stalk, disintegrating with age, thin walled, readily separable from gleba hyphae, nonamyloid (Figs. 1D and 1E); ascospores hyaline to pale yellow, globose, (12) 14–19 (22) μm in diam. excluding the ornamentation (Figs. 1D and 1E); ornamentation of blunt spines connected in an irregular alveolate reticulum, 1–4 μm high, mostly have a de Bary bubble and are uniguttulate, walls 1.5–2 μm thick (Fig. 1F).

In terms of ecology, all collections were found in the vicinity of Populus alba L. ssp. pyramidalis (Bunge) W.Wettst. Soils were sandy to finely sandy with a history of extensive management practices. Soils have relatively high water-soluble salt content (1.29 g kg−1), neutral pH (7.34), containing 1.49 g kg−1 of total nitrogen, 46.4 mg kg−1 of available phosphorus, and 29.82 g kg−1 of organic matter. The average annual precipitation for the collections from the sampled region of Inner Mongolia (area of the Bao Tou City) were more similar to European collections, Mediterranean collections from several countries, and Continental collections from Hungary, than to other Asian collections (Table 1, Fig. 2). The Inner Mongolian collections were from sites with larger winter/summer temperature differences and lower winter rainfall averages when compared to the other collections included in this study (Fig. 2).

Figure 2 Average monthly precipitation and temperature for different area.

Average monthly precipitation (right) and average monthly temperature (left) for the Chinese (Baotou) (black solid line), other Asian (dark grey), and European (light gray) collections of Mattirolomyces terfezioides.

Phylogenetic analysis

Bayesian and ML phylogenetic analyses resulted in a strongly supported, topologically identical phylogenetic tree with a well-supported major clade containing the studied specimen from Inner Mongolia which clustered together with other M. terfezioides from China, Hungary, Italy, and South Africa (Fig. 3).

Figure 3 Phylogram of the genus Mattirolomyces based on the sequence dataset of the complete ITS region with Elderia arenivaga as outgroup.

Bootstrap values (ML)/posterior probabilities (from Bayesian inference) are shown above or beneath individual branches. Only bootstrap values larger than 60 and posterior possibilities over 0.95 are shown. Bar = 2 changes/100 characters.

A phylogenetic network analysis (Fig. 4) separated all the five recognized taxa in the genus Mattirolomyces with an unexpected higher diversity displayed in M. terfezioides. The phylogenetic distance of the outgroup (Elderia arenivaga) from the genus Mattirolomyces indicated a poor yet the most optimal selection regarding the available taxa and their sequences. At the base of the Mattirolomyces cluster, three lineages were disclosed. The first lineage led to three clusters: one directed to South Africa with M. austroafricanus, the second with a more basal position of M. spinosus from south Asia (the collection was from Pakistan), and a phylogenetically close collection from the United States, with a distinct sub-cluster of closely related M. mexicanus from Mexico and distantly-related (based on the comparison of the number of mutated sites) M. mulpu from Australia. For all four mentioned recent taxa, the number of available nucleotide sequences was low. The third lineage formed a cluster of M. terfezioides that showed higher intraspecific diversity and a recognizable geographic pattern with more basal haplotypes from China, followed by collections from S. Korea. At this point of evolution, there was a jump of haplotypes from Asia (China) to Europe (Hungary, Italy, France), and no supported intra-Europe geographic pattern was recognized.

Figure 4 A rooted phylogenetic network of Mattirolomyces based on the sequence dataset of the complete ITS region.

Black dots represent recent taxa, gray dots represent ancestral stages/nodes. The M. terfezioides rDNA ITS sequence obtained in the present study is indicated by a black triangle. Values on mutation vectors represent the number of mutations between two nodes. Names of existing taxa and their geographic origin (country of collections) are given. Phylogenetic network was constructed and tested with a median joining approach.

Discussion

Mattirolomyces terfezioides is a commonly collected edible hypogeous fungus best known for its traditional use in the desert regions of the Southern Hemisphere (Trappe, Kovács & Claridge, 2010). Although it has been used for culinary purposes since ancient Persian Empire (ibid.), its recent distribution outside its optimal ecological zone has not been explored. There are two well-recognized areas of distribution: the Mediterranean and Pannonian basin in Europe (Kagan-Zur et al., 2014; Kovács et al., 2001) and Beijing, along with neighboring regions in China. The taxonomic characteristics of the Inner Mongolian collections fit well in the concept of the Mattirolomyces terfezioides morphological species (Fischer, 1938; Mattirolo, 1887) and also the phylogenetic species (Díez, Manjon & Martin, 2002).

We present the first example of a global phylogenetic network study of the genus Mattirolomyces and its corresponding geographic distribution pattern in M. terfezioides. Phylogenetic networks are known to give a better insight into species ecology and distribution (Fig. 4), an approach frequently used for its ability to visualize evolutionary relationships between nucleotide sequences and depict microevolution events such as the geographical distribution of populations (Huson & Bryant, 2006). The geographic distribution of Mattirolomyces indicates that the origin of the genus was the current Asia-Pacific areas prior to frequent long-distance migration events, one of which brought M. terfezioides to eastern and Northern China. Based on the ecology of recently studied areas, the collections from Inner Mongolia and the Shanxi province were both similar to all European collections and also shared the same node in the phylogenetic networks. This supported the idea that climatic conditions were an important evolutionary drive in this species, and that the European M. terfezioides species were probably originated from the China area in Asia. M. terfezioides haplotypes from Europe appeared to form more terminal leaves on the network, indicating their more recent arrival to this area, which is additionally supported by the unresolved haplotype distribution between two main suitable habitats: Mediterranean areas and the Pannonian basin (Kovács et al., 2001). The observed diversification and lack of any further geographic or ecological microevolutionary structure in the European haplotypes in the more terminal leaves of the network additionally support the theory of M. terfezioides recent arrival to this area and/or lack of evolutionary pressure.

Our collections are originated from the continental steppe areas of Inner Mongolia in China, an area characterized by cold semi-arid climates with hot dry summers and cold winters with little snowfall, classified as BSk according to the Köppen-Geiger climate classification, and where climate-related indicators point towards a severe spatial desertification risk (Spinoni et al., 2015). Presently, the area still experiences little rainfall with a low average yearly precipitation and lower average yearly temperatures and winter extremes (>5 degrees lower) compared to any other known Mattirolomyces terfezioides areas. Our findings indicated that this species survives in dryer and colder conditions, at least outside its fruiting period, than previously reported (Gógán Csorbainé et al., 2008), and is becoming further endangered due to projected future climate changes (He et al., 2019). The same collections also showed ecological discrepancies from other currently known species ecologies. Mattirolomyces terfezioidesis usually found under Robinia pseudoacacia L., but rarely under artificially planted Diospyros kaki Thunb., Prunus avium (L.) L. or diverse families of Leguminosae, Ebenaceae, and Rosaceae in Southern and Central Europe and in Northern China (Fischer, 1938; Bratek et al., 1996; Wang, Liu & Sun, 2017). R. pseudoacacia is native to the Southern Appalachian and Ozark Mountains of the United States (Huntley, 1990), and was introduced to Europe, Asia, Australia, South America, and Africa mainly as an ornamental plant, or was cultivated to revegetate disturbed sites or for agricultural and commercial uses in recent centuries (Keresztesi, 1988). Its mycorrhizal association with Mattirolomyces terfezioides is most likely secondary since phylogeographically basal haplotypes of M. terfezioides originated from Asia and not from North America. All our M. terfezioides were collected close to poplars, including pure poplar plantations. As far as we know, this is the first study that shows that M. terfezioides is potentially associated with Populus alba, a tree species in the family of Salicaceae with a wide distribution in Europe and central Asia (Palancean et al., 2018). Populus spp. are among ectomycorrhizal hosts for M. terfezioides, despite Populus spp. form a dual mycorrhiza with the ratio between ectomycorrhiza and arbuscular mycorrhiza depending on specific soil conditions (Neville et al., 2002). Since the European poplar species are most closely related to the Asian species (Cervera et al., 2005), they could be potential mycorrhizal partner for the European M. terfezioides species.

In addition, climate change, recent industry, urbanization, and other land use conversion factors are threatening the survival of M. terfezioides in the wild, and maybe the reason for its long-lost status in China. Populus spp. have only recently become the dominant species in Northern China and are mainly used for the restoration of degraded arid and semi-arid landscapes, combating desertification, and drought resilience strategies (FAO, 2016). All Populus-planted areas are sites where M. terfezioides has a potential to grow. These areas may also serve to protect and preserve this rare desert fungus, as long as suitable agricultural practices for its cultivation are developed and supported, especially in rural, arid, and semiarid areas. However, M. terfezioides in China, especially among the Mongol people, remains underexploited and would require further ecological and M. terfezioides mycorrhizal synthesis investigations in order to fully develop agricultural practices for its sustainable cultivation. M. terfezioides could become an excellent model not only to develop local economy in rural areas, but also to highlight the importance of non-timber forest-related products in otherwise industrial forest tree plantations.

Conclusion

The first record of M. terfezioides and its distribution in Inner Mongolia outside its current distribution area in China with the lowest known temperature and precipitation averages for this species. Our first attempt at phylogenetic network analysis in the genus Mattirolomyces revealed its geographic origin was in Asia-Pacific areas prior to frequent long-distance migration events. M. terfezioides seems to be originated from Inner Mongolia and the Shanxi province of China in Asia and was subsequently transported to Europe. Exploring M. terfezioides ecology and its potential to grow with poplars also increase its potential for cultivation and consumption in Central and Eastern China. This is a completely underexploited possibility among Mongols in China, and it deserves further ecological and mycorrhizal investigations on M. terfezioides in arid areas of China which should be carried out in the near future.

Supplemental Information

Supplemental Information 1 Ascocarp of Mattirolomyces terfezioloides in situ.

Photo taken at the collection site in the desert areas of Inner Mongolia, China.

Click here for additional data file.

Supplemental Information 2 Nombril attached to the ascocarp – its position and size in relation to the mature ascocarp of Mattirolomyces terfezioloides.

Click here for additional data file.

Supplemental Information 3 Gleba – macroscopic view of mature ascocarp of Mattirolomyces terfezioloides.

Cross section of an ascocarp.

Click here for additional data file.

Supplemental Information 4 Microscopic slide of asci and spores from mature Mattirolomyces terfezioloides ascocarp.

Microscopic slide made in Cotton blue. Asci opaque, containing 8-spores, irregularly shaped. Spores hyaline to pale yellow when native, in Cotton blue staining blue-gray, globose, with ornamentation.

Click here for additional data file.

Supplemental Information 5 Macroscopic slide of spores from mature Mattirolomyces terfezioloides ascocarp.

Microscopic slide made in water. Spores hyaline to pale yellow, globose, 14–19 (22) μm in diameter, ornamentation of spores ornamentation 1–4 μm high, forming blunt spines connected in an irregular alveolate reticulum.

Click here for additional data file.

Supplemental Information 6 Spore with warty, blunt spines (scanning electron micrograph).

scanning electron micrography, show that ornamentation of blunt spines connected in an irregular alveolate reticulum, 1–4 μm high, mostly have a de Bary bubble and are uniguttulate, walls 1.5–2 μm thick.

Click here for additional data file.

Supplemental Information 7 Average monthly temperature (°C) and Average monthly precipitation (mm).

Click here for additional data file.

We thank Dr. Xiaojuan Huang for its help with meteorological data collection.

Additional Information and Declarations

Competing Interests

Author Contributions

DNA Deposition

Data Availability

The authors declare that they have no competing interests.

Jie Wei conceived and designed the experiments, analyzed the data, prepared figures and/or tables, authored or reviewed drafts of the article, and approved the final draft.

Tine Grebenc analyzed the data, prepared figures and/or tables, and approved the final draft.

Xuan Zhang performed the experiments, prepared figures and/or tables, and approved the final draft.

SiMin Xiang performed the experiments, authored or reviewed drafts of the article, and approved the final draft.

Yongjun Fan conceived and designed the experiments, performed the experiments, analyzed the data, prepared figures and/or tables, authored or reviewed drafts of the article, and approved the final draft.

The following information was supplied regarding the deposition of DNA sequences:

The sequence is available at GenBank: MN619773.1.

The following information was supplied regarding data availability:

The raw measurements are available in the Figs. S1–S7 and Table 1.

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
