# Peer review of "Distribution and phylogeography of the genus Mattirolomyces with a focus on the Asian M. terfezioides haplotypes"

_PeerJ, doi:10.7717/peerj.13511_

## Round 0.1 · original submission · Major Revisions

Both reviewers made a thorough review and I am sure that taking into account the suggestions and corrections, some of them made directly in the file, the paper will improve. Moreover, both reviewers and I think that the English needs to reach a professional level.

·

Basic reporting

The structure of the article should conform to an acceptable format of ‘standard sections’ (see our Instructions for Authors for our suggested format). Significant departures in structure should be made only if they significantly improve clarity or conform to a discipline-specific custom.

Figures should be relevant to the content of the article, of sufficient resolution, and appropriately described and labeled.

Experimental design

The investigation must have been conducted rigorously and to a high technical standard. The research must have been conducted in conformity with the prevailing ethical standards in the field.
Methods should be described with sufficient information to be reproducible by another investigator.

Validity of the findings

The conclusions should be appropriately stated, should be connected to the original question investigated, and should be limited to those supported by the results. In particular, claims of a causative relationship should be supported by a well-controlled experimental intervention. Correlation is not causation.

Additional comments

I have reviewed the paper about Mattirolomyces, I think it is correct for publication in this journal, but some modifications indicated in the text must be made.
Presents a new hypothesis for the distribution of this rare species of hypogeal Fungus.

Reviewer 2 ·

Basic reporting

This paper discusses the geographic origin and distribution of the genus Mattirolomyces and more precisely M. terfezioides haplotypes based on ecological and phylogenetic data. The phylogeography concept of this species is here discussed in light of new data from a recent collection of M. terfezioides associated with poplar trees in the Inner Mongolia. In particular, this paper suggests that M. terfezioides was originated from the Asian-Pacific areas (Chinese area) prior to frequent long-distance migration events to Europe. It is important to mention that there is a real lack of works on M. terfezioides from the Asian region (only two papers Ka et al. 2015 and Wang et al. 2017). However, while I think that the subject of the paper is interesting, I have several comments and revisions (detailed below) which should be addressed by authors before this manuscript can be reconsidered for publication by PeerJ.

The manuscript needs extensive revision for language by a native English speaker. Some sections need a complete paraphrasing to ease their reading. I made a first overall revision of the paper language and I corrected and added several scientific and technical terms. You will find all those revisions marked in red on the PDF version of the manuscript file enclosed with my revision comments.

Abstract comments:
Line 27: What does authors mean by basal in the sentence: ….. Other Asian haplotypes were more basal?

Introduction comments:
Line 42: … intriguing geographical distribution: the term intriguing is inappropriate and should be replaced
Lines 43-46: The citations concerning the five Mattirolomyces species need references of their original source papers.
Lines 50-53: This paragraph should be revised; there is no connecting idea between the first sentence “The desert regions… high summer temperatures” and the second one Mattirolomyces spp…since prehistoric times (Trappe et al., 2010)”
Lines 51-53: The meaning of the following sentence: “Mattirolomyces spp.… geographical, botanical, and cultural attributes…prehistoric times (Trappe et al., 2010)” is not clear and need a complete language revision.

Experimental design

The submission define well the research question. However, the Material and methods part lack sufficient information to be reproductible by another investigator.

Material and methods comments:
Lines 102-103: Please precise the number of repetitions which were used for the measurements of asci and spores sizes
Lines 90-92: The meaning of this paragraph is not clear and need to be written differently
The section concerning the determination of soil physical and chemical properties should be moved and placed just after the section Study site and sampling.
Lines 109-111: Please specify the target gene for the PCR amplifications, which is normally according to the result data, is the rDNA ITS region. This section should be also amended with the PCR primer pairs used for the amplification of the ITS region.
As the PCR amplifications were performed according to Wang et al. (2017) and these last authors amplified two regions (ITS and LSU), it will be more clear for readers to describe the PCR protocol used for the amplification of the ITS region.

Validity of the findings

Results comments:
Line 156: Authors state that they collected 32 ascocarps of M. terfezioides from Inner Mongolia; however they studied the DNA from only one specimen? Why the DNA of the others 31 specimens were not studied? Studying a large collection of specimens would support the scientific findings of this study and bring probably more explanations to the phylogeographic pattern of the investigated Mattirolomyces species.
Line 161: The two terms solid and spongy used to describe the gleba of M. terfezioides have opposite meanings. Authors should choose one of the two terms.
Line 163: The compact robust hyphal aggregate represented in the picture 1B is called nombril according to Awamah and Alsheikh (1979). So please replace the term sclerotia by nombril in the text.
Lines 174-175: These two sentences “Sclerotia: attached… the roots. Aroma…pleasantly sweet” should be removed from this section as Authors are describing Microscopic features and the nombril and aroma are considered as morphological criteria; besides that they were already cited in the macroscopic description of M. terfezioides (lines 162-163).
Line 178: Authors qualified the soil pH value which corresponds to 7.34 as slightly basic. This value classifies this soil as a neutral soil. In fact, soils can be classified according to their pH value as: neutral—6.5 to 7.5; alkaline—over 7.5; acidic—less than 6.5, and soils with pH less than 5.5 are considered strongly acidic.
Line 180: Please precise the geographic origin of collections targeted in the table 1 for the comparison of the average annual precipitation.
Lines 190, 191: Please check the figures orders in the figures legends and correct it subsequently in the text
Lines 192-194: The meaning of this section “The high distance…taxa and their sequences” is not clear, please reformulate it.

Discussion comments:
Line 237: Authors should give more specifications about the type of BSk climate classified according to the Köppen-Geiger climate classification. Is it an arid or semi-arid climate zone?
Lines 257-259: The meaning of the following sentence « This is another example… on specific soil conditions (Neville et al., 2019) » is not clear. Please rephrase this section and clarify it by adding more precisions on the host plant concerned and by giving a logical sense to this sentence.

Additional comments

Tables comments:

Table 1:
Please change the title to:
Collection information of M. terfezioides nuclear rDNA ITS sequence generated in the present work and ITS sequences from the genus Mattirolomyces used in this study retrieved from GenBank or UNITE databases. Species names and GenBank accession numbers were supplemented with geographic origin of the collection, climate, potential host plants and other site data, if available.
Please replace the title country and area or origin by geographic origin
Please replace the GenBank reference code by GenBank accession numbers


Figures comments:
Figures legends:
There is a redundancy in the figures legends (Fig. 4 and Fig.5) and an incoherency between the figures legends and the different titles attributed to each figure. This section needs an intense revision.
Please change the Fig.1 legend to:
Macro- and micromorphological characteristics of M. terfezioloides.
A Ascocarps in situ; B Nombril attached to the ascocarp; C Gleba; D, E Asci and spores; F Spore with warty, blunt spines (scanning electron micrograph).
Figs. 2 and 3 legends:
Please change the Fig. 3 legend according to the modifications made on this section in the revised version of the manuscript.
Please add to Fig. 4 legend:
The M. terfezioides rDNA ITS sequence obtained in the present study is indicated by a black triangle.

Annotated reviews are not available for download in order to protect the identity of reviewers who chose to remain anonymous.

---

## Round 0.2 · Minor Revisions

I encourage you to invite an English reviewer for your paper to reach an acceptable English level, also please review carefully that the figures are well cited in text and that they correspond to legends. Change the captions if necessary.

Reviewer 2 ·

Basic reporting

The authors addressed the main concerns from my comments and thus achieved the majority of the necessary revisions. However, there are still additional (minor) comments to make as authors added new text sections in order to revise some paragraphs and which contained grammar and vocabulary errors. Furthermore, authors omitted to revise some sections of the figures and tables legends.

Abstract:
Line 29: Authors still didn’t clarify the meaning of the term “basal” in this sentence which seems inappropriate and I suggest replacing it.
- Authors omitted to add the list of keywords with the requested changes

Introduction:
Line 40: Please remove the comma after 1938.
Line 41: As the authors preferred to start a new sentence in this line, they should add the word “this last”: Mattirolo named this last in 1887 as…
Line 42: Please rewrite the sentence as follow: however, Moreno et al. (2012) confirmed later its separate generic position from the genus Choiromyces.
Line 46: Please delete the word “is an” in the sentence: “belongs to the "is an" ascomycetous desert truffles”.
Line 48: Please replace the word “an” by “a”.
Line 57: Please replace the word “in” by “from”.
Line 65: Please replace the words hunting and harvesting in this sentence: “regular hunting and harvesting” by “hunt and harvest”.

Experimental design

Material and Methods:
Lines 125-129: Please rewrite this paragraph as follow: “The macro-morphological characterization of ascomata was performed using a stereomicroscope (Motic K400, Motic Asia, Hong Kong) following the Mattirolomyces morphological characters description of Kovács et al. (2011). The micro-morphological features were determined on 30 spores and asci using a light microscope (Motic BA410E with a Moticam2506 camera, Motic Asia, Hong Kong). Melzer's reagent…
Lines 138-141: Please rewrite this paragraph as follow:
DNA extraction, PCR amplification of the complete rDNA ITS region using primers ITS1f / ITS4 (White et al. 1990) and Taq PCR Master Mix (Biobasic, Ontario, Canada) as well as sequencing were carried out according to Wang et al. (2017). PCR products were purified…

Results:
Line 201: In the first version of the manuscript, authors stated that the peridium was not clearly differentiated from the gleba and in the new revised version of the paper they mentioned the complete opposite. Was the peridium well differentiated from the gleba or not?

Discussion:
Line 278: Please add the word “are” to the following sentence: “Our collections are originated from the continental…”
Line 279: Please replace the word “of” in this sentence by characterized: “an area characterized by cold semi-arid climates…”
Line 305: Authors cannot claim that Asian Populus spp. are most likely the primary ectomycorrhizal host for M. terfezioides since they didn’t examine the root system of these plants and determine the mycorrhizal type formed between the two partners in the field.

Validity of the findings

no comment

Additional comments

Figures comments:
Figures legends
- There is still an incoherency between the figures legends and the different titles attributed to each figure. According to the figures legends, the figures 2 and 3 are combined in one figure (Figure 2 left and right); however, they are represented in the figures section as two separate figures (Fig.2 and Fig.3). This section should be checked and revised according to the Figures legends.
- The Figures and tables legends were well revised in the section figure and tables legends (lines 411-438); however, the figures and tables titles in the next pages remain with the same old captions and should be corrected following the revised version of the figures and tables legends.
Fig. 3: Please remove the automatic grammar checker under the species names in the phylogenetic tree.

---

## Round 0.3 · accepted · Accept

Thanks for the effort of correcting the last issues, mainly of style. They improved the article very much.